# Patient characteristics and changes in anxiety symptoms in patients with panic disorder: Post-hoc analysis of the PARADIES cluster randomised trial

Tobias Dreischulte[1][☯], Karoline Lukaschek[1][☯], Marietta Rottenkolber[1], Jana Werle[1], Thomas S. Hiller[2], Jörg Breitbart[2], Ulrike Schumacher[3], Christian Brettschneider[4], Jürgen Margraf[5], Jochen Gensichen[1]*, on behalf of the PARADIES study group[¶]

1 Institute of General Practice and Family Medicine, University Hospital, LMU Munich, Munich, Germany, 2 Institute of General Practice and Family Medicine, Jena University Hospital, Jena, Germany, 3 Centre for Clinical Studies, Jena University Hospital, Jena, Germany, 4 Department of Health Economics and Health Services Research, Hamburg Center for Health Economics, University Medical Center Hamburg-Eppendorf, Hamburg, Germany, 5 Mental Health Research and Treatment Center, Ruhr-University Bochum, Bochum, Germany

☯ These authors contributed equally to this work.
¶ Membership of the PARADIES study group is provided in the Acknowledgments.
* jochen.gensichen@med.uni-muenchen.de

**Data Availability Statement:** The data that support the findings of this study are available from the

## Abstract

Anxiety disorders are among the most common mental health problems in primary care. The PARADIES (Patient Activation foR Anxiety DIsordErS) intervention combined elements of cognitive behavioural therapy with case management and has demonstrated efficacy. Our aim was to explore patient characteristics, which may influence the course of anxiety symptoms over a 12 months period. Multiple linear regression was used to quantify associations of baseline characteristics (demographics, clinical parameters, medication use) with changes in anxiety symptoms as measured by the Beck anxiety inventory. Treatment modalities (e.g. adherence to appointment schedules) were considered as confounders. We examined univariate associations between dependent and independent variables before considering all independent variables in a multivariate final model. To find the best model to explain BAI score changes, we performed step-wise selection of independent variables based on Akaike information criteria. We tested for interaction terms between treatment allocation (intervention vs control) and independent variables using the multivariate model. We repeated these analyses in control vs intervention groups separately. From the original trial (N = 419), 236 patients (56.3%) were included. In the multivariate model, receiving the intervention (p<0.001), higher anxiety symptom severity (p<0.001) and longer illness duration at baseline (p = 0.033) were significantly associated with changes in anxiety symptom severity to the better while depression severity at baseline (p<0.001) was significantly associated with changes in anxiety symptoms to the worse. In stratified analyses, the control group showed significant associations between depression symptom severity and illness duration with anxiety symptom changes while baseline severity of anxiety symptoms remained significantly associated with anxiety symptom changes in both groups. A brief primary-care-

PARADIES study consortium but restrictions apply to the availability of these data, which were used under license for the current study, and so are not publicly available. Data are however available upon reasonable request. Data requests may be directed at "Stiftung Allgemeinmedizin - The Primary Health care Foundation" (www.stiftung-allgemeinmedizin. de). Mail: office@stiftung-allgemeinmedizin.de.

**Funding:** The PARADIES-study was funded by the Federal Ministry of Education and Research (Bundesministerium für Bildung und Forschung, BMBF) (Funding no.: 01GY1146).

**Competing interests:** The authors have declared that no competing interests exist.

based exposure training combined with case management is effective in a broad range of patients with panic disorder with/without agoraphobia, including those with longer illness duration and co-existing symptoms of depression at baseline.

## Introduction

Anxiety disorders have a high prevalence, with a 12-month rate of about 18% and lifetime rates of about 30% [1–3]. They are among the most common mental health problems seen in primary care settings, where patients with anxiety disorders are predominantly managed [4]. In fact, for most patients with panic disorder, the general practitioner (GP) is the first, and often the only contact [5]. Patients with anxiety disorders experience lower quality of life and increased rates of health care utilisation [6, 7]. Treatment options in primary care include pharmacotherapy (antidepressants and anxiolytics if required) as well as psychotherapeutic approaches, such as cognitive behavioural therapy (CBT) [8–10]. There is also some evidence that case management improves outcomes, which encompasses continuous monitoring and proactive support for patients as a collaborative effort of primary care teams [11–14].

The PARADIES intervention was a practice team–supported exposure training and combined evidence-based elements of CBT with case management [5, 15]. Adult patients diagnosed with PDA (ICD-10: F41.0 or F40.01) were included, while those with suicidality, psychotic or substance-related disorders, severe physical impairments, pregnancy or ongoing anxiety-specific psychotherapy were excluded. Intervention group patients received a therapy companion book providing psychoeducation, instructions on how to perform exposer-exercises, and exposure log sheets. Over a 23-week period, four structured GP visits were scheduled. The first three visits introduced patients to the CBT elements, while the fourth appointment provided relapse-prevention information. From the second visit, patients were encouraged to independently perform anxiety exposure exercises at least twice weekly. Structured telephone monitoring was carried out by a practice nurse in order to enhance treatment adherence and ensure regular monitoring of anxiety symptoms. Where monitoring results were concerning, GPs could arrange additional patient contacts and/or adapt exercise plans. Patients in the control group received treatment as usual. In both groups, GPs could administer any supplementary treatment (including pharmacotherapy or referrals) at their own discretion.

The intervention was evaluated in a cluster randomised controlled trial, the findings of which have previously been published [5, 16]. Briefly, the trial included 73 general practices (419 patients). As described in more detail elsewhere [5], practices first recruited patient participants over a three months period and were subsequently randomized to provide care as usual or deliver the PARADIES intervention to participating patients under their care (allocation ratio 1:1). Overall, 36 practices (230 patients) were randomised to the intervention arm and 37 practices (189 patients) to the control arm. Clinical endpoints were the clinical severity of the anxiety symptoms as assessed by the Beck Anxiety Inventory (BAI) [17]. The intention to treat analysis found that symptoms of anxiety improved to a significantly greater extent in the intervention group (p = 0.008) with an intergroup difference in the reduction of the BAI score (range: 0–63) of 4.0 points [−6.9; −1.2] at twelve months.

The PARADIES trial has therefore established that, on average, patients with anxiety disorder benefit from a combination of CBT and case management. The aim of this study was to examine variability in the course of anxiety symptom severity at 12 months follow up and to

explore patient characteristics, which may explain that variability. In particular, we investigated as explanatory factors patient demographics (since age, sex and level of education may influence motivation and/or ability to engage with the intervention) [18], anxiety symptom severity and illness duration at baseline, co-morbid depression, multimorbidity and polypharmacy as well as the type and/or number of psychotropic drugs used to treat these (since all of these factors may influence the prognosis of panic disorders) [19].

## Methods

### Study design

Following descriptive analysis of baseline characteristics and patient level changes in BAI scores, we first pooled data from all intervention and control group patients in order to examine associations between patient baseline characteristics and patient level changes in the BAI score while controlling for treatment group status (intervention vs control). Differential effects of baseline characteristics on anxiety symptom changes in control and intervention groups were subsequently examined via interaction terms and in stratified analyses. Through stratification by use vs non-use of medication commonly used in the treatment of anxiety disorders (antidepressants and benzodiazepines), we further explored the influence of baseline symptoms of anxiety and depression.

The Ethics Committee of the Friedrich-Schiller University Jena granted approval of the study protocol on 17 August 2012 (no. 3484–06/12). All participating physicians and patients gave their written informed consent to participate in the study.

### Study population

For the purposes of this study, we included all participants in the PARADIES trial for whom there was complete information on all baseline characteristics of interest (see below) and BAI score changes between baseline and 12 months follow up.

### Dependent and independent variables

As the dependent variable (i.e. the outcome measure of interest), we defined the change in severity of anxiety symptoms measured by the self-administered Beck Anxiety Inventory (BAI) [17] over the 12 months study period. The BAI asks patients to rate how severely affected they had been by 21 typical symptoms of anxiety (total score range 0–63) during the previous week.

In order to identify factors associated with changes in anxiety symptoms, we assessed the following categories of variables:

- patient demographics—age, sex, years of education;

- clinical parameters at baseline—illness duration, depression scale (Patient-Health-Questionnaire (PHQ-9) [20]), multimorbidity (defined as chronic somatic conditions according to Bussche > 3 [21]), Patient Assessment of Chronic Illness Care (PACIC) [22];

- medication at baseline—any antidepressants, any benzodiazepines, psychotropic polypharmacy (defined as two or more of the following medicines taken concomitantly: benzodiazepines, z-drugs, opioids, other hypnotics, antidepressants, antipsychotics), and any polypharmacy (defined as 5 or more medicines taken concomitantly);

Given that general practices and patients had a degree of flexibility in delivering and adhering to the intervention, respectively, it is possible that such factors may modify the effect of the

intervention and if so, distort the relationship between baseline characteristics and treatment response. To control for such effects, we therefore considered the following treatment modalities:

- completion of the intervention as intended (attendance at 4th appointment: yes vs no)

- number of monitoring contacts with the case manager (range: 1–10)

- number of additional contacts with the GP (range: 0–2).

## Statistical analyses

To compare intervention and control groups, the t-test or the Mann-Whitney $U$ test was used for metric variables and the $\chi^2$ or Fisher exact test for categorical variables. For all regression models, the dependent variable (change in severity of anxiety symptoms over the 12 months study period) was calculated as the difference in the Beck-Anxiety-Inventory [BAI] [5] between T0 and T2, where we subtracted individual BAI sum scores at T2 from individual BAI sum scores at T0.

As the primary analysis, we examined associations between independent and dependent variables while controlling for allocation status (intervention vs control), using multivariate linear regression. We initially examined univariate associations before considering all independent variables in a multivariate final model. All continuous independent variables were mean-centred. In order to find the best model for explaining BAI score changes, we added all independent variables into multivariate models and performed step-wise selection (both directions: forward and backward) based on AIC criteria (R package MASS; function stepAIC). We also tested interaction terms between treatment allocations and independent variables. In order to examine, whether and to which extent the effect of baseline characteristics on changes in anxiety symptom severity differed between the intervention and control group, we repeated the analyses for the intervention and control groups separately. Since drug treatment may influence baseline symptoms of anxiety and depression, we additionally repeated the analyses for users and non-users of benzodiazepines or antidepressants separately. All effects are reported as significant at $p < 0.05$. Residual plots (residuals vs. fitted values, Q-Q plot, scale-location plot, and residuals vs leverage values) were used to check the assumptions of the linear regression models. Data were analysed using R version 3.6.3 (https://www.r-project.org/).

## Results

Of the 419 patients included in the PARADIES trial, we included in the analysis all 236 (56.3%) patients with complete information on all independent (baseline characteristics) and dependent variables (difference of BAI scores at baseline and 12 months follow up). Of these, 128 patients and 108 patients were members of the intervention and control group, respectively. The proportions of patients with complete information were similar in the intervention group (55.7%) and in the control group (57.1%).

In the overall sample, the mean (SD) age was 45.3 (13.4) years and the majority (75.4%) were women. Table 1 shows that there were no significant differences between included patients of the intervention and control groups and patients were very similar in terms of demographics and clinical parameters at baseline. With respect to medication, the prevalence of polypharmacy and psychotropic polypharmacy were also similar. In both groups, approximately half of the patients were taking antidepressants at baseline and the median defined daily doses (DDD) of antidepressants did not differ between intervention and control patients.

**Table 1. Descriptive statistics of patient characteristics, GP (practice) characteristics and intervention delivery modalities.**

| | Total (n = 236) | Intervention Group (n = 128) | Control Group (n = 108) | Comparison intervention vs control p-value |
|---|---|---|---|---|
| **Patient demographics** | | | | |
| Age [mean (SD)] | 45.3 (13.4) | 46.0 (13.0) | 44.5 (13.9) | 0.3898 |
| Female | 178 (75.4%) | 93 (72.7%) | 85 (78.7%) | 0.2824 |
| Education time: years [mean (SD)] | 11.1 (3.0) | 11.0 (3.0) | 11.2 (3.0) | 0.6287 |
| **Clinical parameters at baseline** | | | | |
| Illness duration in months [median (Q1-Q3)] | 9 (3–19) | 8 (3–19) | 9 (3–19.5) | 0.7123 |
| Depression scale PHQ-9 T0 [mean (SD)] | 11.2 (5.6) | 11.4 (5.7) | 10.8 (5.5) | 0.4163 |
| Multimorbidity (≥3 chronic conditions) [n] | 123 (52.1%) | 65 (50.8%) | 58 (53.7%) | 0.6543 |
| BAI [mean (SD)] | 27.7 (12.2) | 27.1 (11.8) | 28.5 (12.6) | 0.4068 |
| Patient assessment of chronic illness care (PACIC) [mean (SD)] | 6.4 (2.6) | 6.2 (2.4) | 6.6 (2.7) | 0.2896 |
| **Medication at baseline** | | | | |
| Polypharmacy (≥5 medicines) | 48 (20.3%) | 25 (19.5%) | 23 (21.3%) | 0.7372 |
| Psychotropic polypharmacy (≥2 psychotropic medicines) | 49 (20.8%) | 26 (20.3%) | 23 (21.3%) | 0.8527 |
| Any benzodiazepine | 15 (6.4%) | 11 (8.6%) | 4 (3.7%) | 0.1250 |
| Any antidepressant | 108 (45.8%) | 52 (48.2%) | 56 (51.9%) | 0.0846 |
| Antidepressant DDD (median, IQR) | 1.00 (0.5, 1.5) | 1.00 (0.7, 2.0) | 1.00 (0.5, 1.0) | 0.1218 |
| **Intervention delivery parameters** | | | | |
| Appointment 1 performed | | 123 (96.1%) | - | |
| Appointment 2 performed | | 121 (94.5%) | - | |
| Appointment 3 performed | | 112 (87.5%) | - | |
| Appointment 4 performed | | 96 (75.0%) | - | |
| Telephone contacts [mean (SD)] | | 9.1 (2.1) | | |
| Additional contacts | | | | |
| 0 | | 104 (81.3%) | - | |
| 1 | | 16 (12.5%) | - | |
| 2 | | 3 (2.3%) | - | |

* significant difference between intervention and control group (p<0.05).

BAI: Beck-Anxiety-Inventory; PACIC: Patient Assessment of Chronic Illness Care (categories: 1 (0%) to 11 (100%)); PHQ: Patient Health Questionnaire; DDD: Defined daily dose.

The use of benzodiazepine anxiolytics was rare in both groups, albeit somewhat higher in the intervention vs control group (8.6% vs 3.7%).

With respect to intervention delivery parameters, four intervention group patients had missing information on how many and which of the scheduled appointments had been attended. Almost all intervention group patients (96.1%) attended the first appointment, slightly lower proportions attended the second (94.5%) and third appointment (87.5%) while three quarters (75%) attended the fourth appointment.

## Changes in anxiety symptom severity over the study period

Fig 1 shows the distribution of changes in BAI scores in the intervention and control groups, where positive values reflect improvements and negative values reflect worsening of anxiety symptom severity. In the intervention group, the mean BAI (SD) score fell from 28.2 (12.3) to 17.3 (12.5), while in the control group it fell from 28.2 (12.4) to 22.1 (13.3) at 12 months follow

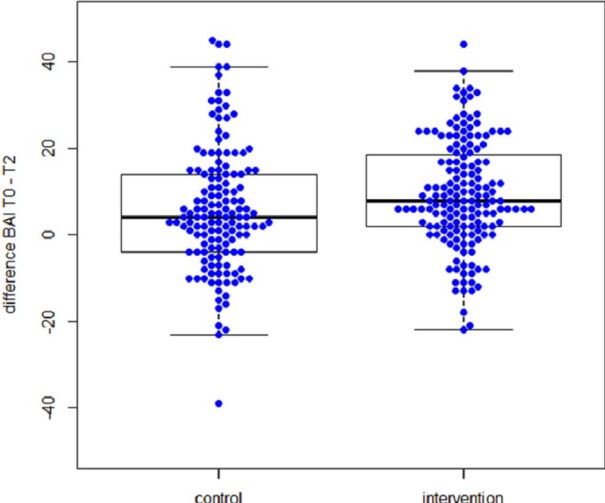

**Fig 1. Distribution of changes in anxiety symptom severity between T0 and T2 among patients allocated to intervention and control groups.** Positive values reflect improvements while negative values reflect worsening of anxiety symptoms.

up. The mean differences in BAI scores were 9.4 (12.4) in the intervention group and 6.0 (14.5) in the control group. In the intervention group, changes in BAI scores ranged from an improvement by 44 points to a worsening by 22 points, and in the control group from an improvement of 44 points to a worsening of 39 points. The majority of patients in the intervention group (n = 103; 80.5%) and in the control group (n = 70; 74.8%) had changes in anxiety symptom severity to the better while 25 (19.5%) intervention group patients and 38 (35.2%) control group patients had no changes in anxiety symptom severity or changes to the worse.

Table 2 compares the proportions of intervention and control group patients by BAI severity category at baseline and at the end of the study period. While there were no significant differences between intervention and control groups at baseline, the groups differed significantly at 12 months follow up. Over the study period, the proportions of patients with severe symptoms decreased from 51.9% to 38.9% in the control group and more than halved (from 52.3% to 22.7%) in the intervention group, while the proportions of patients with no or minimal symptoms increased from 3.7% to 16.7% in the control group and from 1.6% to 28.1% in the intervention group.

### Examination of factors associated with anxiety symptom changes in the whole study population

Tables 3–5 show the findings of linear regression analyses, while model diagnostics and iterative changes in $R^2$ values are provided in the supporting S2 File.

**Table 2. Comparison of intervention and control group patients by BAI severity category at baseline and at the end of the study period.**

| BAI groups (Score range in points) | Baseline | | | End of study | | |
|---|---|---|---|---|---|---|
| | Intervention group | Control Group | p-value | Intervention group | Control Group | p-value |
| No or minimal (0 to 7) | 2 (1.6%) | 4 (3.7%) | 0.2861 | 36 (28.1%) | 18 (16.7%) | 0.0099 |
| Mild (8 to 15) | 25 (19.5%) | 13 (12.0%) | | 32 (25.0%) | 17 (15.7%) | |
| Moderate (16 to 25) | 34 (26.6%) | 35 (32.4%) | | 31 (24.2%) | 31 (28.7%) | |
| Severe (≥26 points) | 67 (52.3%) | 56 (51.9%) | | 29 (22.7%) | 42 (38.9%) | |

**Table 3. Findings of univariate and multivariate linear regression models for all study participants (n = 236).**

| | Univariate regression coefficient (95% CI) | Multivariate regression coefficient (optimal model after the stepwise procedure; 95% CI)# |
|---|---|---|
| Allocated to intervention group (vs treatment as usual) | 4.2 (0.6 to 7.7) * | 5.7 (2.6 to 8.8)*** |
| **Demographics** | | |
| Age [years] | -0.1 (-0.2 to 0.1) | -0.1 (-0.2 to 0.02) |
| Female sex (vs male) | 4.1 (0.1 to 8.2)* | 3.3 (-0.2 to 6.7). |
| Education time [years] | 0.08 (-0.5 to 0.7) | |
| **Clinical parameters at baseline** | | |
| Anxiety symptom severity (BAI T0) | 0.5 (0.4 to 0.7)*** | 0.8 (0.6 to 1.0) *** |
| Illness Duration [months] | 0.01 (-0.1 to 0.1) | 0.2 (0.02–0.4)* |
| Depression scale (PHQ-9) | 0.2 (-0.1 to 0.5) | -0.9 (-1.4 to -0.5)*** |
| Multimorbidity (vs not multimorbid) | -1.0 (-4.5 to 2.6) | |
| Patient assessment of chronic illness care (PACIC) | -0.3 (-1.0.1 to 0.4) | |
| **Medication use at baseline** | | |
| Benzodiazepine (yes vs no) | 3.7 (-3. to 10.9) | 10.4 (-1.3 to 22.0). |
| Antidepressant (yes vs no) | -2.7 (-6.2 to 0.9) | |
| Antidepressant DDD | 0.4 (-2.0 to 2.9) | |
| Polypharmacy ($\geq$5 medicines vs <5) | -1.9 (-6.3 to 2.5) | -3.2 (-7.2 to 0.8) |
| Psychotropic polypharmacy ($\geq$2 psychotropic medicines vs <2) | 1.9 (-2.5 to 6.2) | |
| **Interactions** | | |
| Allocated to intervention group * Anxiety symptom severity (BAI T0) | | -0.3 (-0.6 to 0.005). |
| Allocated to intervention group * Depression scale (PHQ-9) | | 0.8 (0.2 to 1.5)* |
| Allocated to intervention group * Illness Duration [months] | | -0.2 (-0.5 to 0.001). |
| $R^2$ | | 0.3105 |

. <0.1;

* <0.05;

** <0.01;

*** <0.001;

# independent variables multivariate model: group, age, sex, education time, baseline anxiety symptom severity, illness duration, depression scale, multimorbidity, patient assessment of chronic illness care, benzodiazepine, antidepressant, polypharmacy, psychotropic polypharmacy; DDD: Defined daily dose.

Table 3 shows the findings of univariate and multivariate regression analyses for all patients, where allocation status (intervention vs control group) is controlled for. In univariate analysis, three variables were significantly associated with changes in anxiety symptom severity to the better over the study period, namely being allocated to the intervention group (univariate regression coefficient 4.2 (95% CI 0.6 to 7.7); p = 0.020), being female (4.1 (0.1 to 8.2); p = 0.047) and having higher anxiety symptom severity (BAI) at baseline (0.5 (0.4 to 0.7); p<0.001).

In multivariate analysis, receiving the intervention (multivariate regression coefficient 5.7 (95% CI 2.6 to 8.8); p<0.001) and higher anxiety symptom severity at baseline (0.8 (0.6 to 1.0); p<0.001) were still significantly associated with changes in anxiety symptom

**Table 4. Findings of regression analyses stratified by intervention and control status.**

| | Control group | | Intervention group | |
|---|---|---|---|---|
| | Univariate regression coefficient (95% CI) | Multivariate regression coefficient (optimal model after stepwise procedure; 95% CI)# | Univariate regression coefficient (95% CI) | Multivariate regression coefficient (optimal model after stepwise procedure; 95% CI)# |
| **Demographics** | | | | |
| Age [years] | -0.02 (-0.2 to 0.2) | | -0.1 (-0.3 to 0.0) | -0.2 (-0.3 to -0.02)* |
| Female sex (vs male) | 8.1 (1.1 to 15.1)* | 4.5 (-1.3 to 10.3) | 1.9 (-2.8 to 6.6) | |
| Education time [years] | 0.03 (-1.0 to 1.0) | | 0.2 (-0.5 to 0.9) | |
| **Clinical parameters at baseline** | | | | |
| Anxiety symptom severity (BAI T0) | 0.6 (0.4 to 0.8)*** | 0.8 (0.6 to 1.0) *** | 0.4 (0.3 to 0.6)*** | 0.5 (0.3 to 0.6) *** |
| Illness Duration [months] | 0.01 (-0.2 to 0.2) | 0.2 (-0.03 to 0.4)˙ | 0.0 (-0.1 to 0.2) | |
| Depression scale (PHQ-9) | -0.1 (-0.6 to 0.5) | -0.9 (-1.4 to -0.4)*** | 0.4 (0.1 to 0.8)* | |
| Multimorbidity (vs not multimorbid) | -0.5 (-6.4 to 5.4) | | -1.1 (-5.3 to 3.1) | |
| Patient assessment of chronic illness care (PACIC) | -0.4 (-1.5 to 0.7) | | -0.1 (-0.9 to 0.8) | |
| **Medication use at baseline** | | | | |
| Benzodiazepine (yes vs no) | 13.1 (-2.3 to 28.5) | 9.6 (-2.8 to 21.9) | -1.0 (-8.5 to 6.5) | |
| Antidepressant DDD | -0.1 (-4.8 to 4.6) | | 0.7 (-1.9 to 3.4) | |
| Antidepressant (yes vs no) | -4.2 (-10.0 to 1.6) | | -0.5 (-4.8 to 3.8) | |
| Polypharmacy (≥5 medicines vs <5) | -3.9 (-11.1 to 3.2) | -6.0 (-11.8 to -0.2)* | 0.2 (-5.1 to 5.5) | |
| Psychotropic polypharmacy (≥2 psychotropic medicines vs <2) | 3.6 (-3.6 to 10.7) | | 0.5 (-4.7 to 5.8) | |
| **Delivery modalities** | | | | |
| Appointment 4 performed | Not applicable | | 4.7 (-0.4 to 9.9) | |
| Telephone contacts | Not applicable | | 1.0 (-0.04 to 1.9) | |
| Additional contacts | Not applicable | | | |
| 0 | | | Reference | |
| 1 | | | -3.2 (-0.5 to 3.1) | |
| 2 | | | 4.6 (-9.2 to 18.3) | |
| R$^2$ | | 0.3759 | | 0.2117 |

˙<0.1;

*<0.05;

**<0.01;

***<0.001;

# independent variables multivariate model: age, sex, education time, baseline anxiety symptom severity, illness duration, depression scale, multimorbidity, patient assessment of chronic illness care, benzodiazepine, antidepressant, polypharmacy, psychotropic polypharmacy; DDD: Defined daily dose.

severity to the better. In contrast to the univariate analysis, longer illness duration at baseline (0.2 (0.02 to 0.4); p = 0.033) was significantly associated with changes in anxiety symptom severity to the better, whereas more severe symptoms of comorbid depression at baseline (-0.9 (-1.4 to -0.5); p<0.001) were significantly associated with changes in anxiety symptoms to the worse after adjustment for confounding variables. Three interaction terms were also included in the final model indicating differential effects of baseline anxiety symptoms, baseline symptoms of depression and illness duration on changes in anxiety symptom severity in the intervention versus control group. The final multivariate model included 12 variables and three interaction terms and explained 31.1% of variability in anxiety symptom changes (R$^2$ = 0.3105).

**Table 5. Findings of regression analyses stratified by use vs non-use of antidepressants and/or benzodiazepines.**

| | Non-users (n = 120) | | Users (n = 116) | |
| --- | --- | --- | --- | --- |
| | of antidepressants or benzodiazepines | | of antidepressants and/or benzodiazepines | |
| | Univariate regression coefficient (95% CI) | Multivariate regression coefficient (optimal model after the stepwise procedure; 95% CI)# | Univariate regression coefficient (95% CI) | Multivariate regression coefficient (optimal model after the stepwise procedure; 95% CI)# |
| Allocated to intervention group (vs treatment as usual) | 3.0 (-1.8 to 7.7) | 4.8 (0.8 to 8.8)* | 5.0 (-0.2 to 10.3) | 5.8 (1.4 to 10.1)** |
| **Demographics** | | | | |
| Age [years] | -0.1 (-0.2 to 0.1) | | -0.1 (-0.3 to 0.1)· | |
| Female sex (vs male) | -0.5 (-6.0 to 5.1) | | 8.4 (2.5 to 14.3)** | 5.5 (0.4 to 10.6)* |
| Education time [years] | 0.2 (-0.7 to 1.1) | 0.8 (0.0 to 1.5)* | -0.02 (-0.8 to 0.8) | |
| **Clinical parameters at baseline** | | | | |
| Anxiety symptom severity (BAI T0) | 0.6 (0.4 to 0.7)*** | 0.6 (0.4 to 0.8) *** | 0.5 (0.3 to 0.7)*** | 0.7 (0.5 to 0.9) *** |
| Illness Duration [months] | 0.01 (-0.2 to 0.2) | | 0.01 (-0.2 to 0.2) | |
| Depression scale (PHQ-9) | 0.7 (0.3 to 1.1)** | | -0.1 (-0.6 to 0.3) | -0.6 (-1.1 to -0.2)** |
| Antidepressant DDD | | | 4.2 (0.2 to 8.1)* | |
| Multimorbidity (vs not multimorbid) | -1.0 (-5.7 to 3.7) | | -0.8 (-6.2 to 4.5) | |
| Patient assessment of chronic illness care (PACIC) | -0.7 (-1.6 to 0.3) | | 0.1 (-0.9 to 1.1) | |
| **Medication use at baseline** | | | | |
| Polypharmacy (≥5 medicines vs <5) | 3.5 (-2.9 to 10.0)· | 4.4 (-1.0 to 9.9)· | -5.4 (-11.5 to 0.7)· | -7.1 (-12.1 to -2.1)** |
| Psychotropic polypharmacy (≥2 psychotropic medicines vs <2) | 5.1 (-5.7 to 15.8) | | 3.1 (-2.4 to 8.6) | |
| $R^2$ | | 0.3039 | | 0.3429 |

·<0.1;

*<0.05;

**<0.01;

***<0.001;

# independent variables multivariate model: group, age, sex, education time, baseline anxiety symptom severity, illness duration, depression scale, multimorbidity, patient assessment of chronic illness care, benzodiazepine, antidepressant, polypharmacy, psychotropic polypharmacy; DDD: Defined daily dose.

## Examination of factors associated with anxiety symptom changes in intervention vs control patients

Table 4 shows the results of analyses stratified by allocation status (intervention vs control). In the control group, six variables contributed to the final model (female sex, baseline anxiety symptom severity, baseline symptoms of depression, use of benzodiazepines and use of polypharmacy) and explained 38% of variability in anxiety symptom changes ($R^2$ = 0.38). In the intervention group, only two variables (age and baseline anxiety symptom severity) contributed to the final model and explained 21% of variability in anxiety symptom changes ($R^2$ = 0.21).

More severe symptoms of anxiety at baseline were significantly associated with changes in anxiety symptoms to the better in both groups, but the effect was stronger in the control vs intervention group (regression coefficient of 0.8 (0.6 to 1.0)) vs 0.5 (0.3 to 0.7). Only in the intervention group was age significantly associated with changes in anxiety symptom severity to the worse (regression coefficient of– 0.2 (-0.3 to -0.02)). By contrast, only in the control group were more severe symptoms of depression at baseline (regression coefficient of—0.9 (-1.4 to -0.4)) and polypharmacy (regression coefficient of—6.0 (-11.8 to -0.2)) significantly associated with changes in anxiety symptom severity to the worse. Treatment modalities in the

intervention group (i.e. attended appointments, telephone contacts and number of additional unplanned contacts) did not alter any of the effect estimates for the baseline characteristics investigated.

### Examination of factors associated with anxiety symptom changes in users vs non-users of antidepressants or benzodiazepines

Table 5 shows the results of analyses stratified by use vs non-use of antidepressants or benzodiazepines. Again, allocation status and anxiety symptom severity at baseline were significantly associated with anxiety symptom changes in both groups and polypharmacy contributed to both models, whereas education time was relevant only among non-users and female sex only among users of antidepressants or benzodiazepines, respectively. While in both groups, more severe baseline symptoms of anxiety were significantly associated with changes in anxiety symptoms to the better, only among users of antidepressants or benzodiazepines were more severe baseline symptoms of depression significantly associated with changes in anxiety symptoms to the worse.

## Discussion

In this exploratory analysis of anxiety symptom changes in a cohort of patients with panic disorder enrolled in the randomized controlled PARADIES trial, we found considerable variation in anxiety symptom changes ranging from substantial improvements to relevant worsening of symptoms in both intervention and control groups. Symptoms were unchanged or worsened over the 12 months period in 19.5% and 35.2% of intervention and control group patients, respectively. The final multivariate regression model for all patients included 12 variables and explained 31.3% of this variation. Overall, our analyses confirmed the significant intervention effect reported in the pre-specified primary analysis [5]. In analyses stratified by allocation status, six variables contributed to explain variation in anxiety symptoms changes in the control group but only two variables contributed in the intervention group, which re-emphasises that the PARADIES intervention is a decisive driver of anxiety symptom improvements among those receiving it.

Our study found that for patients in both treatment groups, greater anxiety symptom severity at baseline was associated with changes in anxiety symptom severity to the better at 12 months follow up. We could not find other studies which report the changes in symptom severity as the difference in values from measurement at baseline to measurement at follow-up. Generally, the literature so far suggests that patients with more severe symptom severity at baseline tend to have poorer treatment outcomes [23–25], but findings have been mixed for the association of pre-treatment severity with CBT for panic disorder outcomes [26]. On the other hand, a study by Hadjistavropoulos et al. (2016) showed (consistent with our findings) that greater pretreatment condition severity was associated with larger therapy benefits [27]. Given that the latter intervention had higher intensity than the PARADIES trial (12 vs 4 modules of therapist-assisted Internet-delivered cognitive behavior therapy for depression or generalized anxiety), our findings suggest that even lower intensity interventions may benefit patients with more severe symptoms at baseline. We assume that patients who have more severe symptoms, have more potential to improve but may also be more likely to seek and receive treatment and engage with exercises that were part of the intervention [28]. Additionally, our results may partially be explained by regression toward the mean, a statistical effect in which measured values tend to be closer to the population mean than the baseline values due to statistical variability.

A somewhat surprising finding was that longer illness duration was independently associated with changes in anxiety symptoms to the better in the control group (while no such association was found in the intervention group). Seemingly in contrast to our findings, two naturalistic studies have found less favorable outcomes for patients with longer duration of anxiety symptoms without intervention. Ronalds et al. (1997) found that adult patients in general practice with depressive, anxiety or panic disorder (n = 148; DSM-lll-R criteria), more patients showed improvements in anxiety symptom severity when they had an illness duration of less than six months versus six months or more (52% vs 42%) [29]. Penninx et al. (2011) found that in patients with anxiety and/or depression (n = 1209) longer baseline duration of the index disorder was independently associated with lower likelihood of first remission [24]. Nevertheless, the findings of these studies are not directly comparable to ours, since outcomes were measured differently, and in our study, patients were enrolled in a randomised controlled trial. A possible explanation of our findings is that in the control group, patients with longer illness duration may be more likely than those with shorter illness duration to be initiated on effective treatment, whereas all patients in the intervention group were subjected to the same intervention.

It is known that co-morbid anxiety and depression in the same patient negatively affect the clinical course of anxiety symptoms. In previous studies, patients with more severe manifestations of these illnesses typically respond less robustly to treatment than patients with either disorder alone [24, 30, 31]. However, our finding that in stratified analysis, the negative impact of comorbid depression on anxiety symptom severity was limited to the control group, suggests the intervention also yielded a benefit of CBT in patients with more severe symptoms of depression at baseline. This suggests that the intervention is robust to external confounders over 12 months. Moreover, we have previously shown that our short intervention is cost effective regarding total and direct costs as well as disease-specific health care costs [32].

We found medication at baseline not to be significantly associated with anxiety symptom changes in primary analysis (although benzodiazepine use at baseline and polypharmacy were included in the final models for the control group and the study population as a whole). In the case of benzodiazepines, the lack of a significant association may be attributable to the relatively small proportion of study participants using these drugs at baseline. In the case of antidepressants, our findings suggest that actual symptom control of depression is a more important driver for the course of anxiety symptoms than use of antidepressants (which may not sufficiently control depression symptoms).

When we stratified by use vs non-use of benzodiazepines or antidepressants, more severe symptoms of depression at baseline were associated with changes in anxiety symptoms to the worse among users but not among non-users of antidepressants or benzodiazepines at baseline, respectively. This finding suggests that the presence of more severe symptoms of depression in spite of drug treatment marks a group of patients, who may also have a less favorable prognosis of anxiety symptoms.

In terms of demographic variables, female sex was associated with larger changes in anxiety symptoms to the better only in univariate analysis, whereas older age was associated with anxiety symptom changes to the worse among intervention group patients. Previous studies found women to respond more favorably to collaborative care interventions for anxiety and to report a higher commitment to therapy and a stronger belief in the helpfulness of psychotherapy than men [33]. These factors might predict motivation and effort in treatment and also positive clinical outcomes in CBT [34]. Conversely, older age may have the opposite effect [35]. Although female sex was no longer associated with anxiety symptom changes in multivariate analysis, the change in the point estimate was not large (it fell from 4.1 (0.1 to 8.2) to 3.3 (-0.2 to 6.7) in univariate versus multivariate analysis). Our findings can therefore be interpreted as broadly consistent with previous research.

## Strengths and limitations

A major strength of our study is that it is one among very few to examine patient characteristics associated with a more and less favourable course of anxiety symptoms and the effect of intervention. As such it enhances our understanding of who might benefit more and less from the PARADIES intervention, which we consider an essential addition to the primary analysis of intervention effectiveness.

However, our study also faces limitations. First, this is a post-hoc analysis of data from the completed PARADIEs trial and as such it is an exploratory study, which is vulnerable to multiple testing bias. While our findings confirm the findings of the fully powered primary analysis, the identified associations between independent and dependent variables should be interpreted as hypothesis generating rather than hypothesis testing. The sizes of the intervention group (n = 230) and the control group (n = 189) were different due to a higher loss to follow up in the control group, possibly due to the de-blinding of the intervention-status after randomisation. Nevertheless, variables like age, education, depression scale or BAI were well balanced between both groups and all statistical analyses were adjusted for the relevant variables. Second, only 236 of 419 patients (56.3%) could be included in the regression analysis, because BAI was not documented or independent variables were missing. Third, it was not possible to calculate a multilevel model (which is the appropriate method of analysis for cluster-randomized studies) because there were too many practices with only 1–2 patients and our multi-level models did not converge. However, the intraclass correlation coefficient (0.017) was very low in the analysis of the PARADIES trial suggesting minimal cluster effects [5]. Fourth, the final multivariate model only explained approximately a third of the variation in anxiety symptom changes among participants, which suggests there may be additional sources of variation, which we did not measure. Finally, although the intervention focussed on non-pharmacological treatment options, we cannot exclude that differential use of anti-anxiety medication in intervention and control groups during follow up may have contributed to the observed effects.

## Conclusion

Managing patients with panic disorder in primary care is a challenging task. Our study shows that a short intervention combining elements of CBT and case management, and delivered by a joint effort of interdisciplinary staff (GPs and non-medical trained staff) is effective over 12 month in a broad range of patients, particularly among those with more severe symptoms, and including those with longer disease duration and co-existing symptoms of panic disorder and depression. Further research is required to identify effective primary care interventions for a considerable proportion of patients (19.5% in the intervention group of this study) whose symptoms remained stable or worsened over the 12 months study period.

## Supporting information

**S1 File. Pearson correlation coefficient (for continuous variables)/t-test for categorical variables.**
(PDF)

**S2 File. Regression analysis, R2 and model residuals.**
(PDF)

## Acknowledgments

On behalf of the PARADIES study group, the authors would like to thank the participating GPs and patients for their participation.

The authors thank the PARADISE study group for which Jochen Gensichen is lead author: Dr. Wolfgang Blank, Kirchberg im Wald, Germany; Dr. Florian Bleibler, Hamburg-Eppendorf, Germany; Jörg Breitbart, Jena, Germany; Dr. Christian Brettschneider, Hamburg-Eppendorf, Germany; Anne Brokop, Jena, Germany; Prof. Dr. Jochen Gensichen, München, Germany; Thomas Hiller, Jena, Germany; Dr. Heike Hoyer, Jena, Germany; Dr. Bert Huenges, Bochum, Germany; Michelle Kaufmann, Jena, Germany; Prof. Dr. Hans-Helmut König, Hamburg-Eppendorf, Germany; Dr. Armin Mainz, Korbach, Germany; Prof. Dr. Jürgen Margraf, Bochum, Germany; Pauline Masopust, Jena, Germany; Alexander Piwtorak, Jena, Germany; Rebekka Salzmann, Jena, Germany; Prof. Dr. Sylvia Sänger, Gera, Germany; Mercedes Schelle, Jena, Germany; Prof. Dr. Peter Schlattmann, Jena, Germany; Dr. Konrad Schmidt, Jena, Germany; Nico Schneider, Jena, Germany; Dr. Elisabeth Schöne, Jena, Germany; Dr. Sven Schulz, Jena, Germany; Dr. Ulrike Schumacher, Jena, Germany; Dr. Michael Sommer, Jena, Germany; Monika Storch, Jena, Germany; PD Dr. Tobias Teismann, Bochum, Germany; Franziska Theune-Hobbs, Jena, Germany; Dr. Paul Thiel, Hildesheim, Germany; Prof. Dr. Michel Wensing, Heidelberg, Germany.

## Author Contributions

**Conceptualization:** Thomas S. Hiller, Jörg Breitbart, Jürgen Margraf, Jochen Gensichen.

**Data curation:** Ulrike Schumacher, Christian Brettschneider.

**Formal analysis:** Marietta Rottenkolber, Ulrike Schumacher.

**Funding acquisition:** Jochen Gensichen.

**Methodology:** Tobias Dreischulte, Karoline Lukaschek, Marietta Rottenkolber, Jana Werle.

**Project administration:** Jörg Breitbart, Jochen Gensichen.

**Supervision:** Tobias Dreischulte, Karoline Lukaschek, Jürgen Margraf, Jochen Gensichen.

**Visualization:** Marietta Rottenkolber.

**Writing – original draft:** Tobias Dreischulte, Karoline Lukaschek, Marietta Rottenkolber.

**Writing – review & editing:** Tobias Dreischulte, Karoline Lukaschek, Jana Werle, Thomas S. Hiller, Christian Brettschneider, Jürgen Margraf, Jochen Gensichen.

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
