## [Decision Letter · Decision Letter 0]

22 Mar 2022

PONE-D-22-02155Patient characteristics and changes in anxiety symptoms in patients with panic disorder: Post-hoc analysis of the PARADIES cluster randomised trialPLOS ONE

Dear Dr. Lukashek

Thank you for submitting your manuscript to PLOS ONE. After careful consideration, we feel that it has merit but does not fully meet PLOS ONE’s publication criteria as it currently stands. Therefore, we invite you to submit a revised version of the manuscript that addresses the points raised during the review process.

We look forward to receiving your revised manuscript.

Kind regards,

Dr. Burak Yulug

Academic Editor

PLOS ONE

Journal Requirements:

https://journals.plos.org/plosone/s/file?id=ba62/PLOSOne_formatting_sample_title_authors_affiliations.pdf\\

4. One of the noted authors is a group or consortium (PARADIES study group). In addition to naming the author group, please list the individual authors and affiliations within this group in the acknowledgments section of your manuscript. Please also indicate clearly a lead author for this group along with a contact email address.

Reviewers' comments:

Reviewer's Responses to Questions

**Comments to the Author**

1. Is the manuscript technically sound, and do the data support the conclusions?

Reviewer #1: Yes

Reviewer #2: Partly

2. Has the statistical analysis been performed appropriately and rigorously? 

Reviewer #1: Yes

Reviewer #2: No

3. Have the authors made all data underlying the findings in their manuscript fully available?

Reviewer #1: Yes

Reviewer #2: Yes

4. Is the manuscript presented in an intelligible fashion and written in standard English?

Reviewer #1: Yes

Reviewer #2: No

5. Review Comments to the Author

Reviewer #1: This is a secondary regression analysis from an already-completed cluster randomized clinical trial. As such, it is an exploratory study, not powered to detect significant treatment differences, and it should be so-stated in the text. The comment "p<.05 is considered significant" means nothing with no adjustment for many, many hypothesis tests. I suggest a comment that "p-values should be interpreted as a guide, rather than as a determinant of significance". Alternatively, if there are confirmatory hypotheses, these should be stated clearly, power analysis given, and Bonferroni adjustments for multiple tests.

I appreciate that they are concerned about analyzing this data set incorrectly, given they are ignoring the cluster effect, and they do indicate some homogeneity across clusters. Finally, any regression course would require all students to do diagnostics for the model, including residual analyses and graphical inspection. I would expect no less in a scientific paper!

Reviewer #2: Dear Editor,

I am stating the evaluations about the study below.

Kind regards.

1. A large part of the introduction describes the PARADIES study, including the independent variables investigated in the study and their relationship with panic disorder in the introduction section.

2. Out of 236 patients in the study, 128 were in the intervention group and 108 were in the control group. In the method section, it is explained in detail how the patients were assigned to the groups.

3. Comparison of both groups (intervention & control) in terms of sociodemographic characteristics (with t test and Chi-square tests for independent groups) and giving p values in Table 1 in the Results section.

4. Half of the patients are receiving antidepressant treatment at the beginning, comparing the drug doses used by the patients using antidepressants in terms of equivalence (For example, 10 mg. Escitalopram = 20 mg. Citalopram = 20 mg. Fluoxetine = 50 mg. Sertalin = 20 mg. Paroxetine).

5. Of the 128 patients, 44 improved, 22 worsened. Of the 106 patients, 44 improved and 39 worsened. Rates are given on worsening only. Giving rates over recovery. (The rate of 74.8% on page 9 should be 64.8%.)

6. Adding the following table to the results section and making its statistical analysis.

The patients in both the intervention group and the control group were divided into groups according to the scores they got from the Beck Anxiety Inventory at the baseline and at the end of the 12-month follow-up, and comparing the two groups.

0-7 points (No anxiety)

8-15 points (Mild anxiety)

16-25 points (Moderate anxiety)

≥ 26 points (Severe anxiety)

7. Showing the correlation analysis with a table in the results section in terms of the relationship between the independent variables included in the regression analysis and the dependent variable (Pearson correlation coefficients).

8. R2 (R square) changes should be given in multiple regression analysis. It shows how much of the variation in the variance (indicates how much the established model determines the investigated relationship) of the relevant independent variable explains. In this study, the model determined 31.1% of the variance. The share of each independent variable in this percentage should be shown by giving the change in R2 (R square).

9. Table 2 and Table 3 need to be redone. Multiple regression analyzes should be performed separately for the intervention group and the control group, and the discussion section should be rewritten according to the results of this analysis (Univariate regression analysis does not need to be specified).

10. One of the dilemmas of the study is that BDZ (Benzodiazepine) and AD (Antidepressant) treatments used in baseline affect both baseline anxiety severity and depression severity. Therefore, the effect of the intervention applied in the study will be understood more accurately in a sample that does not use drugs. It is recommended to exclude the patients using drugs in both the 128-person intervention group and the 106-person control group, and to analyze the symptoms above for the new intervention and control groups consisting of patients who do not use drugs, and to interpret the similarities and differences between the results found with the 128 and 106-person samples in the discussion section.

6. PLOS authors have the option to publish the peer review history of their article (what does this mean?). If published, this will include your full peer review and any attached files.

Reviewer #1: No

Reviewer #2: **Yes: **Abdullah Burak UYGUR

---

## [Author Response · Author response to Decision Letter 0]

20 May 2022

OUR RESPONSE: The manuscript now meets PLOS ONE's style requirements, including those for file naming.

2) We note that you have indicated that data from this study are available upon request. PLOS only allows data to be available upon request if there are legal or ethical restrictions on sharing data publicly. In your revised cover letter, please address the following prompts:

a. If there are ethical or legal restrictions on sharing a de-identified data set, please explain them in detail (e.g., data contain potentially identifying or sensitive patient information) and who has imposed them (e.g., an ethics committee). Please also provide contact information for a data access committee, ethics committee, or other institutional body to which data requests may be sent.

b. If there are no restrictions, please upload the minimal anonymized data set necessary to replicate your study findings as either Supporting Information files or to a stable, public repository and provide us with the relevant URLs, DOIs, or accession numbers. Please see http://www.bmj.com/content/340/bmj.c181.long for guidelines on how to de-identify and prepare clinical data for publication. For a list of acceptable repositories, please see http://journals.plos.org/plosone/s/data-availability#loc-recommended-repositories.

OUR RESPONSE: The data that support the findings of this study are available from the PARADIES study consortium but restrictions apply to the availability of these data, which were used under license for the current study, and so are not publicly available. Data are however available from the authors upon reasonable request. Data requests may be directed at Jochen.Gensichen@med.uni-muenchen.de

3) We note that you have included the phrase “data not shown” in your manuscript. Unfortunately, this does not meet our data sharing requirements. PLOS does not permit references to inaccessible data. We require that authors provide all relevant data within the paper, Supporting Information files, or in an acceptable, public repository. Please add a citation to support this phrase or upload the data that corresponds with these findings to a stable repository (such as Figshare or Dryad) and provide and URLs, DOIs, or accession numbers that may be used to access these data. Or, if the data are not a core part of the research being presented in your study, we ask that you remove the phrase that refers to these data.

OUR RESPONSE: We now provide this data (univariate association of treatment modalities with BAI changes) in the new table 4.

4) One of the noted authors is a group or consortium (PARADIES study group). In addition to naming the author group, please list the individual authors and affiliations within this group in the acknowledgments section of your manuscript. Please also indicate clearly a lead author for this group along with a contact email address.

OUR RESPONSE: We provide the names of the PARADIES study group at the end of the manuscript in the acknowledgement

Reviewer #1: 

1) This is a secondary regression analysis from an already-completed cluster randomized clinical trial. As such, it is an exploratory study, not powered to detect significant treatment differences, and it should be so-stated in the text. The comment "p<.05 is considered significant" means nothing with no adjustment for many, many hypothesis tests. I suggest a comment that "p-values should be interpreted as a guide, rather than as a determinant of significance". Alternatively, if there are confirmatory hypotheses, these should be stated clearly, power analysis given, and Bonferroni adjustments for multiple tests.

OUR RESPONSE: We agree with the reviewer. We have now added in a number of places that we conducted an exploratory analysis and now state in the discussion under limitations: 

“This is a post-hoc analysis of data from the completed PARADIEs trial and as such it is an exploratory study, which is vulnerable to multiple testing bias. While our findings confirm the findings of the fully powered primary analysis, the identified associations between independent and dependent variables should be interpreted as hypothesis generating rather than hypothesis testing.”

2) I appreciate that they are concerned about analyzing this data set incorrectly, given they are ignoring the cluster effect, and they do indicate some homogeneity across clusters. 

OUR RESPONSE: We agree, an analysis with a multi-level model would be the best method. However, due to the lower number of cases in our analysis (n=236) compared to the main study (n=419), only 1-2 patients remain in many centres. Therefore, an analysis with a multi-level model is no longer possible because the models do not converge. Due to the low intraclass correlation coefficient (0.017) in the main study and the associated low cluster effect, we decided to perform the analysis without multi-level models. We now further explain under limitations that our multi level models did not converge. 

3) Finally, any regression course would require all students to do diagnostics for the model, including residual analyses and graphical inspection. I would expect no less in a scientific paper!

OUR RESPONSE: We performed diagnostics for all models including the residual plots. We have added the following sentence in the methods section "Residual plots (residuals vs. fitted values, Q-Q plot, scale-location plot, and residuals vs leverage values) were used to check the assumptions of the linear regression models". In addition, we added the residual plots for the multivariate models in the supporting file 2.

Reviewer #2: 

1) A large part of the introduction describes the PARADIES study, including the independent variables investigated in the study and their relationship with panic disorder in the introduction section.

OUR RESPONSE: We have now added to the introduction the following paragraph to explain the rationale for the independent variables investigated: In particular, we investigated as influencing factors patient demographics (since age, sex and level of education may influence motivation and/or ability to engage with the intervention (19)), symptoms of other psychiatric co-morbidities and multimorbidity as well as the type and/or number of psychotropic and other drugs used to treat these (since all of these factors may influence the prognosis of panic disorders (20). 

2) Out of 236 patients in the study, 128 were in the intervention group and 108 were in the control group. In the method section, it is explained in detail how the patients were assigned to the groups.

OUR RESPONSE: We have now included the following details describing the randomisation process in lines 78 ff: As described in more detail elsewhere (17), practices first recruited patient participants over a three months time period and were subsequently randomised to provide care as usual or deliver the PARADIES intervention to participating patients under their care (allocation ratio 1:1).

3) Comparison of both groups (intervention & control) in terms of sociodemographic characteristics (with t test and Chi-square tests for independent groups) and giving p values in Table 1 in the Results section.

OUR RESPONSE: We now provide the findings of t test and Chi-square tests and provide p values in Table 1 in the results section. 

4) Half of the patients are receiving antidepressant treatment at the beginning, comparing the drug doses used by the patients using antidepressants in terms of equivalence (For example, 10 mg. Escitalopram = 20 mg. Citalopram = 20 mg. Fluoxetine = 50 mg. Sertalin = 20 mg. Paroxetine).

OUR RESPONSE: For antidepressant users at baseline, we have calculated the defined daily dose of antidepressants taken and compared them between groups. The data is presented in table 1. We have also included this variable in linear regression analyses and provide the findings in tables 3 to 5.

5) Of the 128 patients, 44 improved, 22 worsened. Of the 106 patients, 44 improved and 39 worsened. Rates are given on worsening only. Giving rates over recovery. (The rate of 74.8% on page 9 should be 64.8%.)

OUR RESPONSE: We think there is a confusion between patients and changes in BAI points, which we hope to clarify by the following rewording: In the intervention group, changes in BAI scores ranged from an improvement by 44 points to a worsening by 22 points, and in the control group from an improvement of 44 points to a worsening of 39 points. We additionally provide the numbers (%) of patients who improved (had changes to the better) and those who had no changes or had changes to the worse. 

6) Adding the following table to the results section and making its statistical analysis.

The patients in both the intervention group and the control group were divided into groups according to the scores they got from the Beck Anxiety Inventory at the baseline and at the end of the 12-month follow-up, and comparing the two groups.

0-7 points (No anxiety)

8-15 points (Mild anxiety)

16-25 points (Moderate anxiety)

≥ 26 points (Severe anxiety)

OUR RESPONSE: We agree that the manuscript benefits from this additional analysis. We have added the findings in a new table 2 and summarised them in text as follows: Table 2 compares the proportions of intervention and control group patients by BAI severity category at baseline and at the end of the study period. While there were no significant differences between intervention and control groups at baseline, the groups differed significantly at 12 months follow up. Over the study period, the proportions of patients with severe symptoms decreased from 51.9% to 38.9% in the control group and more than halved (from 52.3% to 22.7%) in the intervention group, while the proportions of patients with no or minimal symptoms increased from 3.7% to 16.7% in the control group and from 1.6% to 28.1% in the intervention group.

7) Showing the correlation analysis with a table in the results section in terms of the relationship between the independent variables included in the regression analysis and the dependent variable (Pearson correlation coefficients).

OUR RESPONSE: We provide the findings of the correlation analysis in a supporting file 1. We suggest that the findings do not substantially add to the univariate relationships between independent and dependent variables reported in the manuscript and are therefore dispensable in the main manuscript. 

8) R2 (R square) changes should be given in multiple regression analysis. It shows how much of the variation in the variance (indicates how much the established model determines the investigated relationship) of the relevant independent variable explains. In this study, the model determined 31.1% of the variance. The share of each independent variable in this percentage should be shown by giving the change in R2 (R square).

OUR RESPONSE: In order to find the multivariate model that best explains the change in BAI, we used a stepwise procedure based on the Akaike Information Criteria (AIC). The AIC is similar to the adjusted R squared measure and penalizes when more variables are included in the model. The result of this procedure is a simple model with the optimal number of variables. If present, multicollinearity is also removed by this procedure. 

We provide R2 changes in the supporting file 2.

9) Table 2 and Table 3 need to be redone. Multiple regression analyzes should be performed separately for the intervention group and the control group, and the discussion section should be rewritten according to the results of this analysis (Univariate regression analysis does not need to be specified).

OUR RESPONSE: We have carefully considered this comment and would like to propose that we keep the analysis for the study population as a whole (where we adjust for allocation status), but additionally provide the findings of analysis stratified by intervention and control group. This approach maximises power for examining explanatory variables and avoids missing differential effects in intervention and control groups. We have applied this approach in the revised manuscript and made amendments to methods, results and discussion accordingly.

10) One of the dilemmas of the study is that BDZ (Benzodiazepine) and AD (Antidepressant) treatments used in baseline affect both baseline anxiety severity and depression severity. Therefore, the effect of the intervention applied in the study will be understood more accurately in a sample that does not use drugs. It is recommended to exclude the patients using drugs in both the 128-person intervention group and the 106-person control group, and to analyze the symptoms above for the new intervention and control groups consisting of patients who do not use drugs, and to interpret the similarities and differences between the results found with the 128 and 106-person samples in the discussion section.

OUR RESPONSE: We have further stratified the analysis by use of antidepressants or benzodiazepines and provide the findings in the manuscript and made amendments to methods, results and discussion accordingly. We have opted for this approach of adjusting for allocation status rather than a four way stratification by both intervention and control groups because this would have left few individuals in each stratum, which would have limited power even further.

---

## [Decision Letter · Decision Letter 1]

7 Jul 2022

PONE-D-22-02155R1Patient characteristics and changes in anxiety symptoms in patients with panic disorder: Post-hoc analysis of the PARADIES cluster randomised trialPLOS ONE

Dear Dr. Lukaschek,

Thank you for submitting your manuscript to PLOS ONE. After careful consideration, we feel that it has merit but does not fully meet PLOS ONE’s publication criteria as it currently stands. Therefore, we invite you to submit a revised version of the manuscript that addresses the points raised during the review process.

Reviewers' comments:

Reviewer #1: All comments have been addressed

Reviewer #2: Dear Authors,

The article has some structural problems and this situation continues despite the revision. The independent variables researched in the introduction part are not included enough. Some parts that should be included in the method are included in the introduction.

It is a serious problem that the study did not include only panic disorder patients. In addition, the fact that both groups were treated with medication is a situation that may seriously affect the results of the study, and the daily dose of antidepressant parameter added to the table by the researchers does not relieve these reservations. In addition, the allocation status, which is at the center of the research, is a binary variable. Although sometimes binary variables are treated as independent variables, classically, dependent and independent variables should be numerical in regression analysis. In this case, it is confusing about the variable in the center of the research. Based on the above-mentioned reasons, it was deemed appropriate to reject the article.

We look forward to receiving your revised manuscript.

Kind regards,

Burak Yulug

Academic Editor

PLOS ONE

Reviewers' comments:

Reviewer's Responses to Questions

**Comments to the Author**

1. If the authors have adequately addressed your comments raised in a previous round of review and you feel that this manuscript is now acceptable for publication, you may indicate that here to bypass the “Comments to the Author” section, enter your conflict of interest statement in the “Confidential to Editor” section, and submit your "Accept" recommendation.

Reviewer #1: All comments have been addressed

Reviewer #2: (No Response)

2. Is the manuscript technically sound, and do the data support the conclusions?

Reviewer #1: (No Response)

Reviewer #2: Partly

3. Has the statistical analysis been performed appropriately and rigorously? 

Reviewer #1: (No Response)

Reviewer #2: N/A

4. Have the authors made all data underlying the findings in their manuscript fully available?

Reviewer #1: (No Response)

Reviewer #2: Yes

5. Is the manuscript presented in an intelligible fashion and written in standard English?

Reviewer #1: (No Response)

Reviewer #2: Yes

6. Review Comments to the Author

Reviewer #1: (No Response)

Reviewer #2: Dear Authors,

The article has some structural problems and this situation continues despite the revision. The independent variables researched in the introduction part are not included enough. Some parts that should be included in the method are included in the introduction.

It is a serious problem that the study did not include only panic disorder patients. In addition, the fact that both groups were treated with medication is a situation that may seriously affect the results of the study, and the daily dose of antidepressant parameter added to the table by the researchers does not relieve these reservations.

In addition, the allocation status, which is at the center of the research, is a binary variable. Although sometimes binary variables are treated as independent variables, classically, dependent and independent variables should be numerical in regression analysis. In this case, it is confusing about the variable in the center of the research.

Based on the above-mentioned reasons, it was deemed appropriate to reject the article.

Best regards.

7. PLOS authors have the option to publish the peer review history of their article (what does this mean?). If published, this will include your full peer review and any attached files.

Reviewer #1: No

Reviewer #2: No

---

## [Author Response · Author response to Decision Letter 1]

29 Jul 2022

PONE-D-22-02155R1

Patient characteristics and changes in anxiety symptoms in patients with panic disorder: Post-hoc analysis of the PARADIES cluster randomised trial

PLOS ONE

1) The article has some structural problems and this situation continues despite the revision. The independent variables researched in the introduction part are not included enough. Some parts that should be included in the method are included in the introduction.

Our response: This study is in an exploratory post hoc analyses of data from the PARDIES trial, and the findings of the primary analysis have been published previously. In our opinion, describing the PARADIES intervention and findings of the trial is essential background information for the reader to understand what the objectives of this paper are. We believe the aim of the analysis is clearly stated, namely “to examine variability in the course of anxiety symptom severity at 12 months follow up and to explore patient characteristics, which may explain that variability.”

With regards to describing the independent variables and its rationale in the introduction, we have added additional information in our previous revision upon the reviewer’s request and have made further specifications in the revised manuscript as follows: “In particular, we investigated as explanatory factors patient demographics (since age, sex and level of education may influence motivation and/or ability to engage with the intervention)[19], anxiety symptom severity and illness duration at baseline, co-morbid depression, multimorbidity and polypharmacy as well as the type and/or number of psychotropic drugs used to treat these (since all of these factors may influence the prognosis of panic disorders) [20].” 

We suggest that providing this level of detail in the introduction should be sufficient for the reader to appreciate the rationale for our selection of independent variables. We provide further details on these variables and how they were defined in the methods section.

2a) It is a serious problem that the study did not include only panic disorder patients. 

Our response: We would like to clarify that all included patients had panic disorder. We agree that in patients with comorbid depression and panic disorder, it is more difficult to isolate effects on either condition. 

However, anxiety and depression share clinical symptoms and causes due to genetic pleiotropy and share psychological, social, and neurobiological risk mechanisms. As a result, comorbidity of anxiety and depression is the rule rather than an exception. For instance, of the individuals with a primary depression diagnosis in the Netherlands Study of Depression and Anxiety (NESDA), 67% had a current and 75% had a lifetime comorbid anxiety disorder diagnosis. Similarly, of those with a primary anxiety disorder diagnosis, 63% had a current and 81% a lifetime depressive disorder diagnosis. Thus, limiting the trial population to those with anxiety disorder but without symptoms of depression would have substantially compromised the applicability/external validity of the findings. 

The PARADIES trial has collected data on both depression and anxiety symptoms and all multivariate analyses in this paper took baseline depression symptom severity into account. Our findings suggest that depression symptom severity may mark a group of patients with a less favourable prognosis (as the analysis of control group patients shows) who may nevertheless be at least partially responsive to the PARADIES intervention (as our analysis of intervention group patient shows).

• Choi, K. W., Kim, Y. K., & Jeon, H. J. (2020). Comorbid anxiety and depression: clinical and conceptual consideration and transdiagnostic treatment. Anxiety Disorders, 219-235.

• Demyttenaere, K., & Heirman, E. (2020). The blurred line between anxiety and depression: hesitations on comorbidity, thresholds and hierarchy. International Review of Psychiatry, 32(5-6), 455-465.

• Groen, R.N., Ryan, O., Wigman, J.T., Riese, H., Penninx, B.W., Giltay, E.J., ... & Hartman, C.A. (2020). Comorbidity between depression and anxiety: assessing the role of bridge mental states in dynamic psychological networks. BMC medicine, 18(1), 1-17.

• Hirschfeld R.M.A. The comorbidity of major depression and anxiety disorders: recognition and management in primary care. Prim Care Companion J Clin Psychiatry. 2001;3(6):244–54.

• Lamers F., van Oppen P., Comijs H.C., Smit J.H., Spinhoven P., Van Balkom A.J.L.M., et al. Comorbidity patterns of anxiety and depressive disorders in a large cohort study: the Netherlands Study of Depression and Anxiety (NESD A). J Clin Psychiatry. 2011;72(3):341–8.

• Ter Meulen, W. G., Draisma, S., van Hemert, A. M., Schoevers, R. A., Kupka, R. W., Beekman, A. T., & Penninx, B. W. (2021). Depressive and anxiety disorders in concert–A synthesis of findings on comorbidity in the NESDA study. Journal of affective disorders, 284, 85-97

• Penninx, B., Pine, D., Holmes, Reif, A. Anxiety disorders, Lancet. 2021 March 06; 397(10277): 914–927.

2b) In addition, the fact that both groups were treated with medication is a situation that may seriously affect the results of the study, and the daily dose of antidepressant parameter added to the table by the researchers does not relieve these reservations. 

Our response: We agree that including patients with and without anti-anxiety medication introduces heterogeneity in the study population. However, we adjusted for medication use (including dose as requested by the reviewer in his/her last review) at baseline in all analyses. In order to minimise any residual confounding, the reviewer recommended in his/her last review “to exclude the patients using drugs in both the 128-person intervention group and the 106-person control group, and to analyse the symptoms above for the new intervention and control groups consisting of patients who do not use drugs, and to interpret the similarities and differences between the results found with the 128 and 106-person samples in the discussion section.” In our previous revision, we therefore further stratified the analysis by use of antidepressants or benzodiazepines and provide and discuss the findings in the manuscript.

In the current version of the manuscript, we have added the following sentence to the strengths and limitations section in order to address further concerns regarding the medication: “Finally, although the intervention focussed on non-pharmacological treatment options, we cannot exclude that differential use of anti-anxiety medication in intervention and control groups during follow up may have contributed to the observed effects.”

• Gensichen, J., Hiller, T. S., Breitbart, J., Brettschneider, C., Teismann, T., Schumacher, U., Lukaschek, K., Schelle, M., Schneider, N., Sommer, M., Wensing, M., König, H. H., Margraf, J., & Jena-PARADISE Study Group (2019). Panic Disorder in Primary Care. Deutsches Arzteblatt international, 116(10), 159–166. https://doi.org/10.3238/arztebl.2019.0159

• Gensichen, J., Hiller, T. S., Breitbart, J., Teismann, T., Brettschneider, C., Schumacher, U., Piwtorak, A., König, H. H., Hoyer, H., Schneider, N., Schelle, M., Blank, W., Thiel, P., Wensing, M., & Margraf, J. (2014). Evaluation of a practice team-supported exposure training for patients with panic disorder with or without agoraphobia in primary care - study protocol of a cluster randomised controlled superiority trial. Trials, 15, 112. https://doi.org/10.1186/1745-6215-15-112

3) In addition, the allocation status, which is at the center of the research, is a binary variable. Although sometimes binary variables are treated as independent variables, classically, dependent and independent variables should be numerical in regression analysis. In this case, it is confusing about the variable in the center of the research. 

Our response: We agree with the reviewer that the dependent variable in the linear regression models used must be a metric variable. This requirement is met in all models in the manuscript. However, this requirement does not apply to independent variables. Binary variables can also be considered in regression models as independent variables in addition to metric variables. For example, in most regression models the binary variable "sex" is included as an independent variable. Hence, the inclusion of the variable "allocation status" as an independent variable in the linear regression models is correct from our point of view. Furthermore, the aim of the publication is "to examine variability in the course of anxiety symptom severity at 12 months follow up and to explore patient characteristics, which may explain that variability". Therefore, the objective is not only to show a possible correlation of the change in anxiety symptoms with the variable "allocation status", but also a possible correlation with other "patient characteristics”.

---

## [Editor Report · Decision Letter 2]

20 Sep 2022

Patient characteristics and changes in anxiety symptoms in patients with panic disorder: Post-hoc analysis of the PARADIES cluster randomised trial

PONE-D-22-02155R2

Dear Dr. Lucaschek,

We’re pleased to inform you that your manuscript has been judged scientifically suitable for publication and will be formally accepted for publication once it meets all outstanding technical requirements.

Kind regards,

Burak Yulug

Academic Editor

PLOS ONE

Additional Editor Comments (optional):

I have carefully checked your revised paper and agree that major improvements required for the publication has been done. 
---

## [Editor Report · Acceptance letter]

21 Sep 2022

PONE-D-22-02155R2 

Patient characteristics and changes in anxiety symptoms in patients with panic disorder: Post-hoc analysis of the PARADIES cluster randomised trial 

Dear Dr. Lukaschek:

I'm pleased to inform you that your manuscript has been deemed suitable for publication in PLOS ONE. Congratulations! Your manuscript is now with our production department. 

Kind regards, 

on behalf of

Dr. Burak Yulug 

Academic Editor

PLOS ONE